# Nasopharyngeal carriage of *Streptococcus pneumoniae* among Brazilian children: Interplay with viral co-infection

**Kauana Pizzutti**[1]*, **Juliana Comerlato**[1,2], **Daniele Vargas de Oliveira**[1], **Amanda Robaina**[1], **Mariana Preussler Mott**[1], **Pedro Uriel Pedrotti Vieira**[1], **Tiago Fetzner**[1], **Gabriela Rosa da Cunha**[1], **Muriel Primon de Barros**[1], **Jaqueline Verardo**[1], **Neide Maria Bruscato**[3,4], **João Carlos Batista Santana**[5], **Roberta Rigo Dalla Corte**[5], **Emilio Hideyuki Moriguchi**[3,5], **Vlademir Vicente Cantarelli**[1], **Cícero Armídio Gomes Dias**[1]

1 Federal University of Health Sciences of Porto Alegre, Porto Alegre, RS, Brazil, 2 Hospital Moinhos de Vento, Porto Alegre, RS, Brazil, 3 Moriguchi Institute, Veranópolis, RS, Brazil, 4 Community Hospital São Peregrino Lazziozi, Veranópolis, RS, Brazil, 5 Hospital of Health Clinic of Porto Alegre, Porto Alegre, RS, Brazil

* pizzuttik@gmail.com

**Data Availability Statement:** All relevant data are within the manuscript and its Supporting Information files

## Abstract

Nasopharyngeal transmission of *Streptococcus pneumoniae* is a prerequisite for the development of pneumococcal diseases. Previous studies have reported a relationship between respiratory viruses and *S. pneumoniae* infections. However, there are few studies on this issue among healthy children. This study aimed to examine the relationships between these agents in healthy children from Southern Brazil. This cohort study included 229 nasopharyngeal samples collected from children aged 18–59 months at baseline. *S. pneumoniae* was detected using bacterial culture, whereas respiratory viruses were identified using quantitative polymerase chain reaction. A questionnaire was used at the time of sample collection and medical records were reviewed 14 days after participant inclusion. The prevalence of pneumococcal carriage was 63.7% (146/229), while respiratory viruses were detected in 49.3% (113/229) of the children. Respiratory viruses were more frequently found among pneumococcal carriers than among non-carriers (54.4% vs. 39.7%, p = 0.033). Additionally, rhinovirus (hRV) was more frequent among the pneumococcal carriers (39% vs. 21.7%, p = 0.012), and the presence of human bocavirus (hBOV) alone was associated with the absence of pneumococcal carriage (2.7% vs. 10.8%, p = 0.016). No differences were found in the frequency of pneumococcal carriage, respiratory virus detection, or the co-occurrence of clinical symptoms and diagnosis in the participants 14 days after specimen collection. Our findings revealed a positive relationship between pneumococcal carriage and respiratory virus detection, particularly for hRV. However, we did not observe a relationship between nasopharyngeal respiratory viruses and pneumococci detection during medical appointments, respiratory symptoms, or diseases. This study was one of the first investigations in Latin America to explore the relationship between respiratory viruses and pneumococcal carriage in a healthy children.

**Funding:** The author(s) received no specific funding for this work.

**Competing interests:** The authors have declared that no competing interests exist.

## Introduction

*Streptococcus pneumoniae* (pneumococcus) is a common cause of invasive diseases, such as pneumonia, meningitis, and bacteremia. Since the global introduction of pneumococcal conjugate vaccines (PCV), the incidence of invasive pneumococcal disease (IPD) has declined [1–4]. However, this disease remains among the leading causes of morbidity and mortality in children under five years old and the elderly, particularly in low- and middle-income countries [5]. Global estimates indicate that pneumococcus caused 8.9 million cases of pneumonia in children aged 1–59 months and 294,000 deaths among children aged 1–59 months worldwide in 2015 [6]. Although conjugate vaccines are available, Brazil continues to experience a high annual pneumonia rate, primarily based on the number of hospitalizations for pneumonia in children during their first year of life, reaching 3,433 per 100,000 inhabitants [7]. Pneumococcal vaccine coverage in Brazil in 2017 was approximately 92%, with rates reaching 93.7% in southern Brazil [8].

Nasopharyngeal pneumococcal colonization (PNC) is a prerequisite for pneumococcal disease and facilitates transmission within the community [9, 10]. Various factors influence pneumococcal carriage, including age, nursery attendance, number of siblings, and vaccination [11, 12]. Viruses may also affect the prevalence of pneumococcal colonization. Recent studies have reported an increase in pneumococcal density when respiratory viruses were simultaneously present in the nasopharynx [13–16]. Increased density promotes pneumococcal invasion and the severity of IPD [12]. However, no specific value was established to reliably predict the possibility of pneumococcal pneumonia [17–20].

These viruses primarily cause acute respiratory infections (ARI) during childhood. During viral infection, PNC is associated with increased viral activity and progression to ARI [20–23]. Interactions between respiratory viruses and pneumococci increase the severity of respiratory illnesses [13, 14, 21, 24]. Additionally, the role of various viruses in creating conditions for bacterial superinfections, such as pneumococcal pneumonia, was previously documented [25, 26]. Epidemiological studies have shown that an increase in certain viruses among children is associated with higher rates of pediatric admission owing to IPD [24, 26, 27]. Furthermore, mechanistic studies have revealed that viruses increase bacterial adhesion, translocation, and persistence [28–31].

The relationship between viral infections and pneumococcal diseases has been thoroughly examined. However, there information regarding the interaction between respiratory viruses and pneumococcal carriages in healthy children is lacking [14, 15]. Several studies have explored the heightened severity of infectious diseases when these pathogens interact [13–16, 21, 31]. However, there is a notable gap in data concerning children who are carriers of both pneumococcal and viral pathogens. This study aimed to examine the relationship between respiratory viruses, pneumococcal carriage, and the development of respiratory diseases in Brazilian children aged 18–59 months.

## Materials and methods

### Study population and data collection

We conducted a cohort study to investigate pneumococcal carriage and respiratory viruses in the nasopharynx of children aged 18–59 months. The participants were recruited between February 2018 and October 2019 from Veranópolis City, Rio Grande do Sul, Brazil. The city had an estimated population of 26,533 inhabitants in 2017, 781 of whom were children aged 18–59 months old. This age group included children who had complete vaccination schedules. Participants were recruited through schools, telephone calls, and radio announcements in the city.

All legal guardians provided written informed consent at the time of specimen collection using an informed consent form. If siblings were included, the legal guardian provided consent for both children. All participants were considered healthy enough to attend school and maintain a normal routine at the time of data collection. Individuals were deemed ineligible for enrollment if they had been diagnosed with acute respiratory disease or had been hospitalized in the week preceding specimen collection at enrollment. Health professionals trained for the study collected a nasopharyngeal specimen and conducted structural interviews with legal guardians to collect data on demographic characteristics, pneumococcal vaccination history, and the presence of symptoms, such as cough, coryza/congestion, or sore throat. The children did not undergo medical examinations.

The medical records of each participant were accessed in November 2019 and analyzed for up to 14 days after collection. This period was selected because respiratory disease and/or symptoms following co-infection with pneumococcus and respiratory viruses appear to manifest within a period of 1 to 2 weeks [15, 16, 20–22]. Data were retrieved from 15 different health services covering the entire city to identify symptoms and relevant outcomes. Symptoms, including cough, nasal congestion/coryza, sore throat, and fever, were considered. These outcomes were categorized according to the clinical syndromes: community-acquired pneumonia, sinusitis, acute otitis media, tonsillopharyngitis, asthma, and upper and lower ARI. All relevant outcomes were recorded in the database, with each occurrence counted only once per child, even if the same outcome was presented multiple times to the same participant. If there were no records of visits for a child, it was assumed that the child did not seek medical care for outcomes of interest during that period. The diagnoses documented in the records were based on clinical criteria established by each medical professional. Subsequently, each case was thoroughly reviewed by a pediatric researcher in our team (JCS).

## Respiratory sample collection, and testing

Nasopharyngeal specimen collection followed the Specimen Collection Guidelines–CDC protocol [32], utilizing FLOQswab® (516CS01. Copan®, Murrieta, CA, USA), which were immediately placed in vials containing 1.0 mL skim milk, tryptone, glucose, glycerol transport medium and stored on dry ice. All nasopharyngeal specimens were sent to the Federal University of Health Sciences of Porto Alegre. Subsequently, the specimens were transferred to a freezer at -80˚C until culture. Pneumococcal isolation and identification were conducted using a previously described protocol [32], involving broth enrichment, followed by growth on a blood agar base.

The viral nucleic acid was extracted from NP specimens using the MagMAX™ Pathogen RNA/DNA Kit (Applied Biosystems® by Life Technologies, Carlsbad, CA, USA) following the manufacturer's recommendations. The *18S* rRNA gene was selected as an internal control to monitor nucleic acid extraction efficiency and potential PCR inhibition [33]. Ten respiratory viruses were screened: respiratory syncytial virus, influenza viruses (INF) A and B, metapneumovirus (MPV), human rhinovirus (hRV), adenovirus (ADV), human parainfluenza virus (hPIV) 1–3, and human bocavirus (hBOV). The quantitative PCR assays used are detailed in S1 Table and have been previously described [34–36].

## Ethical approval

The study was approved by the Ethics Committees of Federal University of Health Sciences of Porto Alegre (approval numbers 2.176.785 and 3.374.087) and the Federal University of Rio Grande do Sul (Hospital de Clínicas de Porto Alegre; approval numbers 2.106.235 and 3.063.051).

## Statistical analysis

The sample size was calculated considering the Brazilian PNC prevalence of 62.3% [37], with a 95% confidence level (CI) and an acceptable error of 5.3%, resulting in a required sample size of 229 children. Categorical variables were analyzed using frequencies and percentages, whereas continuous variables were summarized using means and standard deviations. The association between pneumococcal carriage and/or respiratory virus detection and age was assessed using the Kruskal–Wallis test followed by post-hoc Dunn's pairwise tests and Bonferroni correctionThe association of pneumococcal carriage with demographic data, symptoms, medical record analysis, and respiratory viruses was assessed using chi-square or Fisher's exact tests. Odds ratios and multinomial logistic regression, where applicable. The multinomial logistic regression included four outcome categories including 'agent identified' (reference group), 'pneumococcal carriage only, ' 'respiratory virus detection only,' and 'co-occurrence.' The exposure variables included age (months), male sex, lack of sleep, season, and any illness symptoms at enrollment. A chi-square or Fisher's exact test was carried out on the symptoms at the time of collection, and each virus was investigated using two cut-off points: Ct <30 (high levels only) and <35 (any level); Ct >35 was considered negative. Results with a *p-value* < 0.05 were considered statistically significant. Statistical analyses were performed using SPSS software (IBM SPSS Statistics for Windows, Version 25.0. Armonk, NY: IBM Corp.).

## Results

In 2017, Veranópolis City had an estimated 781 children aged 18–59 months. A total of 229 children were included in this cross-sectional study (Fig 1). Five children were excluded from the medical record evaluation, resulting in 224 children included in the cohort study. The study population comprised equal numbers of females and males. Siblings were not clustered in the analyses because of the small number of participants under this condition (only 10 participants).

The prevalence of pneumococcal carriage was 63.7% (146/229, CI 95%: 57.4–69.8%) and 49.3% (113/229, CI 95%: 42.9–55.8%) for respiratory viruses. Respiratory viruses were more frequently found among pneumococcal carriers (54.4% (80/146) vs. non-carriers [39.7% (33/83); p = 0.033]). CI95%: 46.7–62.7%). Co-occurrence rate of 34.9% (80/229, CI 95%: 29.0–41.3%). As shown in Table 1, children who tested positive for viral infection, with or without the co-occurrence of pneumococcus, were younger than pneumococcal carriers (31 and 33 months vs. 42 months; p < 0.001). There was a significant relationship between viral and bacterial co-occurrence and male sex (63.8%; p = 0.004), whereas the absence of the investigated organisms was associated with females (Table 1).

Additionally, multinomial logistic regression analysis of certain demographic data and the presence of pneumococcal carriage, respiratory virus, and co-occurrence was conducted (S2 Table). With each passing month of age, the likelihood of acquiring a respiratory virus decreased by 4% (p = 0.020; OR 0.96, 95% CI 0.92; 0.99), and 5% for co-occurrence (p = 0.003; OR 0.95, 95% CI 0.92; 0.98). The male sex was associated with a higher co-occurrence rate (p = 0.001; OR 3.62, 95% CI 1.64; 7.98). The likelihood of co-occurrence was higher in fall (p = 0.018; OR 5.3, 95% CI 1.33; 21.09), compared with that in spring, which had the highest number of children without microorganisms detected. The symptoms associated with each virus at the time of collection were investigated using two cutoffs: Ct <30 (only high levels) and <35 (any level), as detailed in Table 2. The only symptom associated with the occurrence of any virus was sore throat with any hRV Ct value (p < 0.05).

Various viruses were detected in children with pneumococcal infections. A relationship was observed between total hRV (alone + co-detection) 39% (57/146, p = 0.011) and the

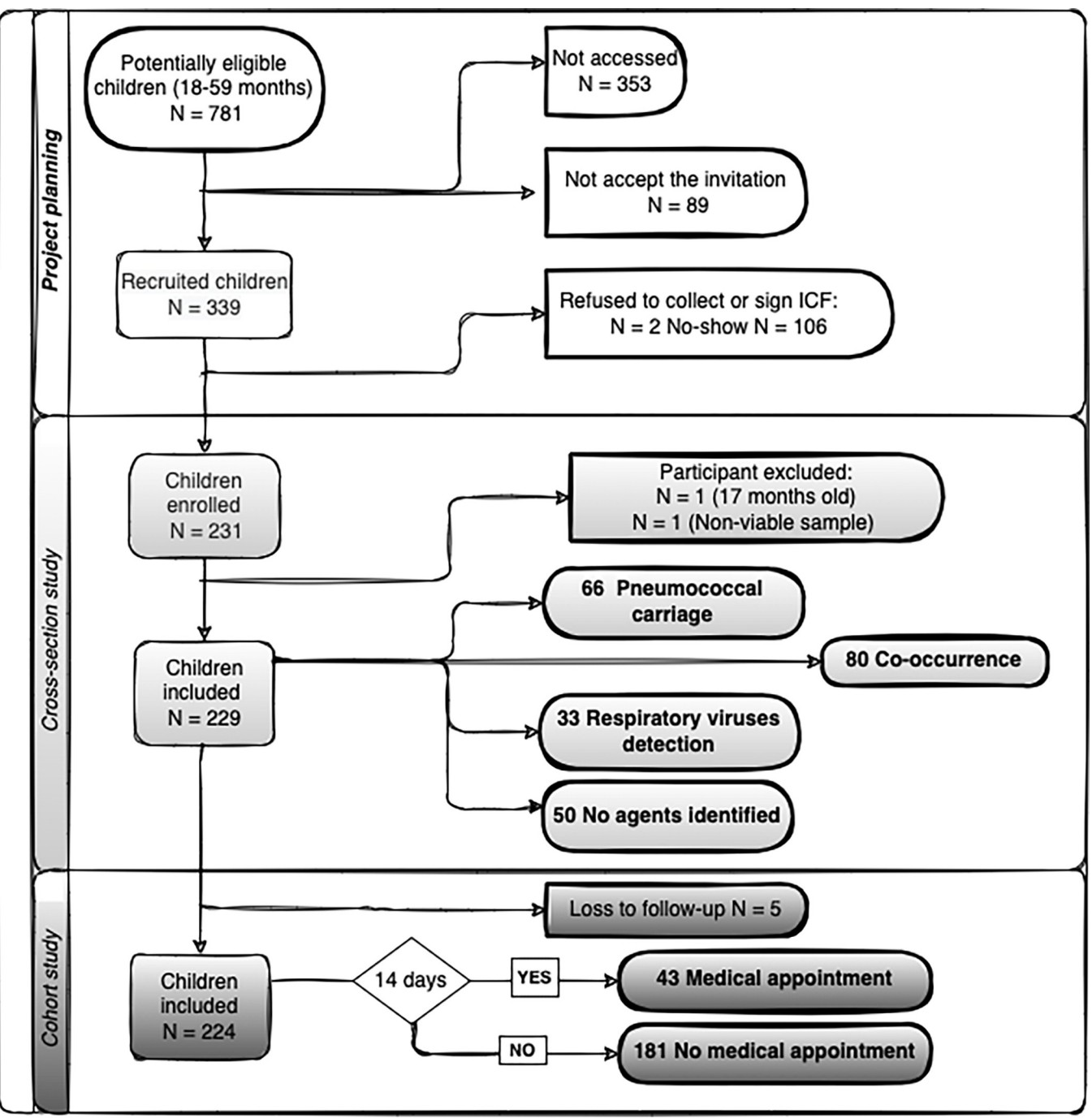

**Fig 1. Flow diagram of children aged 18–59 months, Veranópolis/RS, Brazil, between 2018 and 2019.** ICF, informed consent form.

detection of any respiratory virus 54.8% (80/146, p = 0.040) with pneumococcal carriage (Table 3). Additionally, the presence of hBOV alone was associated with the absence of pneumococcal carriage (10.8%; p = 0.016). All samples tested negative for INF A and B, as well as for hPIV 1, 2, and 3.

**Table 1. Demographic characteristics of the population (n = 229) according to identified microorganisms, Veranópolis/RS, Brazil, 2018–2019.**

| | Total | Only Pneumococcal carriage | Only Respiratory viruses detection | Co-occurrence [3] | No agents identified | *p-value* |
|---|---|---|---|---|---|---|
| **n, (%)** | 229 | 66 (28.8%) | 33 (14.4%) | 80 (34.9%) | 50 (21.8%) | |
| **Age months, median [IQR]** | 36 [27; 48] | 42 [30; 51] | 31 [25; 47] | 33 [24; 43] | 42 [29; 54] | <0.001[4] |
| **Male Sex** | 115 (50.2%) | 30 (45.5%) | 17 (51.5%) | 51 (63.8%) | 17 (34%) | **0.008** |
| Female Sex | 114 (49.8%) | 36 (54.5%) | 16 (48.5%) | 29 (36.2%) | 33 (66%) | - |
| Self-reported race | | | | | | 0.428 |
| White | 212 (92.5%) | 58 (87.9%) | 31 (93.9%) | 77 (96.3%) | 46 (92%) | |
| Black | 2 (0.9%) | 0 (0%) | 0 (0%) | 1 (1.3%) | 1 (2%) | |
| Mixed race | 13 (5.7%) | 7 (10.6%) | 1 (3%) | 2 (2.5%) | 3 (6%) | |
| Indigenous | 1 (0.4%) | 1 (1.5%) | 0 (0%) | 0 (0%) | 0 (0%) | |
| Doesn't know | 1 (0.4%) | 0 (0%) | 1 (3%) | 0 (0%) | 0 (0%) | |
| **Asthma** | 13 (5.7%) | 3 (4.5%) | 3 (9.1%) | 4 (5%) | 3 (6%) | 0.812 |
| **Diabetes** | 0 (0%) | 0 (0%) | 0 (0%) | 0 (0%) | 0 (0%) | - |
| **Kidney Disease** | 2 (0.9%) | 0 (0%) | 0 (0%) | 1 (1.3%) | 1 (2%) | 0.629 |
| **Heart Disease** | 1 (0.4%) | 0 (0%) | 0 (0%) | 0 (0%) | 1 (2%) | 0.309 |
| **Liver Disease** | 1 (0.4%) | 0 (0%) | 0 (0%) | 0 (0%) | 0 (0%) | 0.600 |
| **Cystic Fibrosis** | 0 (0%) | 0 (0%) | 0 (0%) | 0 (0%) | 0 (0%) | - |
| **Sickle cell Anemia** | 2 (0.9%) | 1 (1.5%) | 0 (0%) | 1 (1.3%) | 0 (0%) | 0.759 |
| **Cancer** | 0 (0%) | 0 (0%) | 0 (0%) | 0 (0%) | 0 (0%) | - |
| **Vaccine status** [1] | 225 (98.3%) | 65 (98.5%) | 31 (93.9%) | 80 (100%) | 49 (98%) | 0.169 |
| **School attendance** | 213 (93%) | 61 (92.4%) | 31 (93.9%) | 77 (96.3%) | 44 (88%) | 0.347 |
| **More than one child < 5 years old lives in the same household** | 32 (14%) | 6 (9.1%) | 7 (21.2%) | 11 (13.8%) | 8 (16%) | 0.404 |
| **Other children< 5 years old sleep in the same bedroom of the participant** | 25 (10.9%) | 9 (13.6%) | 6 (18.2%) | 5 (6.3%) | 5 (10%) | 0.248 |
| **Other people sleep in the same bedroom of the participant** | 163 (71.2%) | 41 (62.1%) | 27 (81.8%) | 60 (75%) | 35 (70%) | 0.167 |
| **Family has a member smoker** | 46 (20.1%) | 10 (15.2%) | 10 (30.3%) | 18 (22.5%) | 8 (16%) | 0.266 |
| **Education levels of legal guardians** | | | | | | 0.801 |
| Elementary school | 38 (16.6%) | 13 (19.7%) | 7 (21.2%) | 12 (15%) | 6 (12%) | |
| High school | 80 (34.9%) | 19 (28.8%) | 11 (33.3%) | 31 (38.8%) | 19 (38%) | |
| Undergraduate/graduate | 111 (48.5%) | 34 (51.53%) | 15 (45.5%) | 37 (46.3%) | 25 (50%) | |
| **Breastfeed** | | | | | | |
| Yes | 215 (93.9%) | 61 (92.4%) | 30 (90.9%) | 76 (95%) | 48 (98%) | 0.488 |
| Information unavailable | 1 (0.4%) | 0 (0%) | 0 (0%) | 0 (0%) | 1 (2%) | |
| **Breastfeeding at the time of interview (n = 212)** [2] | 32 (15.1%) | 6 (10%) | 7 (23.3%) | 12 (16%) | 7 (14.9%) | 0.415 |
| **Any illness symptoms at the interview** | 131 (57.2%) | 42 (63.6%) | 18 (54.5%) | 40 (50%) | 31 (62%) | 0.337 |
| Cough | 79 (34.5%) | 22 (33.3%) | 12 (36.4%) | 26 (32.5%) | 19 (38%) | 0.918 |

*(Continued)*

**Table 1.** (Continued)

| | Total | Only Pneumococcal carriage | Only Respiratory viruses detection | Co-occurrence [3] | No agents identified | *p-value* |
|---|---|---|---|---|---|---|
| Nasal congestion/coryza | 108 (47.2%) | 37 (56.1%) | 14 (42.4%) | 34 (42.5%) | 23 (46%) | 0.347 |
| Sore throat | 5 (2.2%) | 0 (0%) | 0 (0%) | 4 (5%) | 1 (2%) | 0.158 |
| **Seasons of the year of sample collection** | | | | | | 0.146 |
| Summer | 6 (2.6%) | 2 (3%) | 1 (3%) | 1 (1.3%) | 2 (4%) | |
| Fall | 102 (44.5%) | 30 (45.5%) | 10 (30.3%) | 45 (56.3%) | 17 (34%) | |
| Winter | 97 (42.4%) | 25 (37.9%) | 19 (57.6%) | 30 (37.5%) | 23 (46%) | |
| Spring | 24 (10.5%) | 9 (13.6%) | 3 (9.1%) | 4 (5%) | 8 (16%) | |

**p<0.05 significant** by Kruskal–Wallis + Bonferroni test (Age) or the chi-square (other variables)

[1] At least one dose PCV 10 and/or PCV13 vaccination.

[2] 17 legal guardian didn't answer if the child was still being breastfeeding

[3] Co-occurrence = Pneumococcal carriage and respiratory virus detection.

[4] Children who tested positive for virus infection and co-occurrence were younger than those that were pneumococcal carriers only

Among the eligible participants for the cohort study, 19.2% (43/224) required medical care within 14 days. Co-occurrence was present in 32.5% (14/43) of children who needed medical care. However, no statistically significant correlation was found between pneumococcal carriage, respiratory virus detection, or co-occurrence in the medical appointments of the participants 14 days after specimen collection (Table 4).

No relationships were found between the frequency of pneumococcal carriage, respiratory virus detection, or co-occurrence and different clinical symptoms and diagnoses of the participants 14 days after the collection of specimens (Table 4). The frequency of each respiratory virus and the symptoms of the children after 14 days of medical care are detailed in S4 Table.

**Table 2. Frequency of clinical symptoms in children with detection of respiratory viruses by real-time PCR using the cut-off levels cycle threshold (CT), Ct- value of < 30 (only high levels) and < 35 (any level), Veranópolis/RS, Brazil, between 2018 and 2019.**

| | Total | RNV | | | ADV | | | BOV | | | RSV | | | MPV | | | Any respiratory virus | | |
|---|---|---|---|---|---|---|---|---|---|---|---|---|---|---|---|---|---|---|---|
| Cycle Threshold (Ct) | | <30 | <35 | > 35 | <30 | <35 | > 35 | <30 | <35 | >35 | <30 | <35 | >35 | <30 | <35 | >35 | <30 | <35 | >35 |
| n, (%) | 229 | 33 (14.4) | 75 (32.8) | 154 (67.2) | 9 (3.9) | 23 (10) | 206 (90) | 6 (2.6) | 27 (11.8) | 202 (88.2) | 3 (1.3) | 17 (7.4) | 212 (92.6) | 2 (0.9) | 17 (7.4) | 212 (92.6) | 47 (20.5) | 113 (49.3) | 116 (50.7) |
| Cough | 79 (34.5) | 11 (33.3) | 26 (34.7) | 53 (34.4) | 2 (22.2) | 5 (21.7) | 74 (35.9) | 0 (0) | 8 (29.6) | 71 (35.1) | 0 (0) | 4 (23.5) | 75 (35.4) | 0 (0) | 5 (29.4) | 74 (34.9) | 13 (27.7) | 38 (33.6) | 41 (35.3) |
| Nasal congestion/ coryza | 108 (47.2) | 17 (51.5) | 35 (46.7) | 73 (47.4) | 3 (33.3) | 10 (43.5) | 98 (47.6) | 1 (16.7) | 10 (37) | 98 (48.5) | 0 (0) | 5 (29.4) | 103 (48.6) | 1 (50) | 7 (41.2) | 101 (47.6) | 21 (44.7) | 48 (42.5) | 60 (51.7) |
| Sore throat | 5 (2.2) | 2 (6.1) | **4 (5.3)** | 1 (0.6) | 0 (0) | 0 (0) | 5 (2.4) | 0 (0) | 0 (0) | 5 (2.5) | 0 (0) | 0 (0) | 5 (2.4) | 0 (0) | 0 (0) | 5 (2.4) | 2 (4.3) | 4 (3.5) | 1 (0.9) |

**p<0.05 significant** by Chi-square or Fisher's exact test

ct >35 was negative

**Table 3. Frequency of respiratory viruses in children with and without pneumococcal carriage, Veranópolis/RS, Brazil, between 2018 and 2019.**

|  | Respiratory virus detection | Pneumococcal carriage | | p-value |
|---|---|---|---|---|
|  |  | Yes (n = 146) | No (n = 83) |  |
| **hRV** | Alone (n = 51) | 38 (26%) | 13 (15.7%) | 0.100 |
|  | Total[1] (n = 75) | 57 (39%) | 18 (21.7%) | **0.011** |
| **ADV** | Alone (n = 9) | 6 (4.1%) | 3 (3.6%) | 1.000 |
|  | Total[1] (n = 23) | 18 (12.3%) | 5 (6%) | 0.195 |
| **hBOV** | Alone (n = 13) | 4 (2.7%) | 9 (10.8%) | **0.016** |
|  | Total[1] (n = 27) | 16 (11%) | 11 (13.3%) | 0.761 |
| **RSV** | Alone (n = 8) | 7 (4.8%) | 1 (1.2%) | 0.264 |
|  | Total[1] (n = 17) | 14 (9.6%) | 3 (3.6%) | 0.163 |
| **MPV** | Alone (n = 2) | 1 (0.7%) | 1 (1.2%) | 1.000 |
|  | Total[1] (n = 17) | 13 (8.9%) | 4 (4.8%) | 0.384 |
| **Any respiratory virus** (n = 113) | | 80 (54.8%) | 33 (39.7%) | **0.040** |

The chi-square or Fisher's exact test

p < 0.05 significant

[1] Total = alone + co-detection with another respiratory virus (S3 Table).

Seasonality could not be reliably determined because of the lack of homogeneity in the sample collection across seasons, as detailed in S5 Table. Samples were collected based on the availability of volunteers, with fewer collections occurring during summer, likely because of school vacations.

**Table 4. Incidence of medical appointments, clinical symptoms and diagnosis in 14 days of medical record analysis according to identified microorganism, Veranópolis/RS, Brazil, between 2018 and 2019.**

|  | Total | Only Pneumococcal carriage | Only Respiratory viruses detection | Co-occurrence[1] | No agents identified |
|---|---|---|---|---|---|
| n, (%) | 224 | 65 (29%) | 31 (13.8%) | 80 (35.7%) | 48 (21.4%) |
| Medical appointments | 43 | 11 (16.9%) | 8 (25.8%) | 14 (17.5%) | 10 (20.8%) |
| Any documented clinical symptoms within 14 days, n (%) | 41 (18.3%)[3] | 11 (16.9%) | 7 (22.6%) | 14 (17.5%) | 9 (18.8%) |
| Cough | 24 (10.71%) | 7 (10.8%) | 5 (16.1%) | 6 (7.5%) | 6 (12.5%) |
| Nasal congestion/ coryza | 18 (8.04%) | 7 (10.8%) | 2 (6.5%) | 4 (5%) | 5 (10.4%) |
| Fever | 13 (5.8%) | 5 (7.7%) | 2 (6.5%) | 2 (2.5%) | 4 (8.3%) |
| Sore throat | 8 (3.6%) | 4 (6.2%) | 0 (0) | 4 (5%) | 0 |
| Any diagnosis within 14 days, n (%) | 43 (19.2%) | 11 (16.9%) | 8 (25.8%) | 14 (17.5%) | 10 (20.8%) |
| Pneumonia | - | - | - | - | - |
| Tonsillopharyngitis | 8 (3.6%) | 6 (9.2%) | 1 (3.2%) | 1 (1.3%) | - |
| Sinusitis | 6 (2.7%) | 1 (1.5%) | 2 (3.2%) | 2 (2.5%) | 2 (4.2%) |
| AOM[2] | 2 (0.9%) | 2 (1.5%) | 0 | 1 (1.3%) | - |
| ARI | 23 (10.3%) | 4 (6.2%) | 4 (12.9%) | 8 (10%) | 7 (14.6%) |
| Asthma | - | - | - | - | - |
| Gastroenteritis | 8 (3.6%) | 1 (1.5%) | 2 (6.5%) | 3 (3.8%) | 2 (4.2%) |

Not statistically significant by the chi-square or Fisher's exact test.

[1] Co-occurrence = Pneumococcal carriage and respiratory virus detection

[2] AOM = acute otitis media

[3] No symptoms were reported at two medical appointments

## Discussion

This study represents an initial investigation in Latin America to explore the relationship between respiratory viruses and pneumococcal infections in healthy children. We found a significant relationship between respiratory viruses, particularly rhinoviruses and pneumococci in children aged 18–59 months in Southern Brazil. However, no association was found with disease development 14 days after exposure to these agents. The prevalence of respiratory viruses was higher in younger children, a finding consistent with previous literature reporting a high viral prevalence in children younger than our study population (< 1 year) [38, 39]. Although our study did not include children under one year of age, we observed a similar relationship between younger age and respiratory viruses in children aged 18–59 months.

Children who were "too sick" were excluded from this study. The recruitment process allowed for the inclusion of children with mild respiratory symptoms as reported by their guardians. Consequently, 108 (47.2%) and 79 (34.5%) children exhibited coryza/congestion and coughing, respectively. Despite the frequency of these symptoms in our study, they were not associated with the detection of the viruses studied. According to the literature, these symptoms are commonly associated with respiratory viruses. Notably, lower Ct levels are often associated with more severe respiratory symptoms and mortality [40]. However, in our study, high Ct levels were observed, which is consistent with the clinical condition of the study group (considered healthy) and the lack of association with symptoms.

An association between hRV and sore throat was identified independent of Ct values, and some studies have demonstrated an association between hRV and sore throat [41, 42]. For instance, in a study conducted on individuals experiencing sore throats, hRV was the most commonly detected pathogen [43]. Moreover, an experimental study involving intranasal inoculation of hRV observed scratchy/sore throat symptoms appearing 10–12 h after the induced inoculation [44]. Notably, these symptom observations were not derived from a medical examination; however, some subjectivity was involved. In the present study, INF and hPIV were not detected. The prevalence of these viruses is typically low (<3%) among asymptomatic children [15, 16] because they interact with their hosts and produce symptoms [45, 46].

No relationship was observed between a specific virus and pneumococcal colonization. However, the detection of more than one virus, especially hRV, is associated with pneumococcal carriage. These findings are consistent with those of Demuri (2018) [15] and Howard (2019) [16], who also reported a relationship between the presence of respiratory viruses and pneumococcal colonization in samples with more than one respiratory virus. In both cited studies, hRV was the most frequent virus involved in the co-occurrence of pneumococcal carriage and significantly influenced the increase in pneumococcal density [15, 16]. There is considerable diversity among studies regarding the age ranges of healthy children. For instance, one study focused on children under three years old [15], while another included children aged four to seven years [16]. Additionally, a study conducted in the Democratic Republic of Congo in children aged 2–60 months reported pneumococcus and viral co-occurrences in 30% of the study population. However, this study did not analyze the co-occurrence of pneumococci and each virus individually [47]. Furthermore, some studies have reported an association between pneumococcus and hRV infection in healthy children. A study conducted in western Australia demonstrated a positive correlation between hRV and pneumococcal carriage in children older than one year [48].

Currently, there is no evidence suggesting an antagonistic relationship between hBOV and pneumococcal carriage. Co-infection with hBOV and pneumococcus has been reported [49–51]. A Brazilian study investigating hBOV-related community-acquired pneumonia in children caused found that pneumococcus was the main bacteria in co-infection cases, with

pneumococcus detected in 33% (7/21) of co-infections [52]. Similarly, a Japanese study observed a positive correlation between the co-detection of pneumococcus and hBOV, where 85.7% (12/14) of children tested positive exclusively for both pathogens in otitis media fluid. However, this correlation was not observed in the nasopharynx, as the proportion of pneumococcus was similar between groups of individuals positive and negative for hBOV and other pathogenic bacteria. Notably, pneumococci were the only pathogens present in all but one of the co-detections in hBOV-positive cases, whereas in the nasopharynx, it was mixed with other pathogenic bacteria [53]. This suggests that these pathogens migrated to their target site (otitis media fluid), with the nasopharynx not being the site where the etiological agent was identified. It is essential to highlight that in children without serious diseases, these microorganisms may play an antagonistic role in the colonization of the nasopharynx, potentially suggesting the protection of pneumococcus in the presence of hBOV alone. However, interpretation of the results of both studies was limited by the small number of patients included. Nevertheless, these studies collectively underscore the relevance of pneumococcus and respiratory virus coinfection in disease pathogenesis. Our findings, such as the absence of INF and hPIV in the sampled population, likely reflect the predominance of these viruses in children with more severe symptoms. INF, in particular, is well known for its association with co-infection and increased risk of subsequent severe pneumococcal disease [54, 55]. Further studies are needed to elucidate the mechanisms underlying the interactions between hBOV and pneumococci.

We examined the clinical outcomes of the population 14 days after specimen collection to gain a deeper insight into the relationships between these microorganisms. We focused on medical consultations, but exclusively recorded the symptoms and diagnoses of interest. The importance of viral infections in predisposing individuals to subsequent bacterial infections is well-established. Among healthy children, studies have demonstrated that the presence of respiratory viruses and pneumococci at high densities is a risk factor for subsequent infections [15, 16, 21]. However, our results suggest harmonious coexistence between viruses and pneumococci. Notably, this population had high vaccination coverage for the prevention of pneumococcal diseases. Therefore, the protection conferred by the vaccine may have prevented the development of disease, regardless of the presence of viral agents. In populations with insufficient vaccination coverage, the relationship with the presence of viruses may differ and warrants further investigation.

These observations suggest that pneumococcal carriage is affected by various factors that are poorly understood, with the participation of viruses likely to play a prominent role. Studies investigating the relationship between respiratory viruses and pneumococcal colonization should be approached with caution. There is considerable diversity across studies, including variations in the age of the children studied, specimen collection methods, microbiological and molecular techniques used for pneumococcal detection, viruses studied, and the definition of outcomes (presence or density of pneumococci). Although this diversity may hinder direct comparisons among study results, it underscores the need for additional research that can offer new perspectives on this issue.

This study had unique characteristics and limitations that merit consideration. This is one of the pioneering investigations analyzing the co-occurrence of pneumococcus and various respiratory viruses among children aged 18–59 months, a population that is predominantly vaccinated against pneumococcal diseases. A limitation of our study, similar to other studies, is the use of PCR to detect respiratory viruses in healthy children. Interpretation of the results can be challenging, as detection may indicate prolonged shedding of the virus resulting from a previous episode [16, 56]. Furthermore, although comprehensive, our viral panel did not

include other agents that could potentially contribute to pneumococcal colonization, such as seasonal coronaviruses.

In summary, our findings revealed a positive relationship between pneumococcal carriage and multiple respiratory viruses, particularly rhinoviruses, in a population with high vaccination coverage. However, despite this association, there was no increased risk of subsequent development of respiratory disease during the 14-day follow-up period. This field of investigation remains open to further studies that may offer insights into this complex issue.

## Supporting information

**S1 Table. Primers and probes used in qPCR assays.**
(PDF)

**S2 Table. Socio-demographic factors associated with only pneumococcal carriage, only respiratory viruses detection, co-occurrence in 229 children, Veranópolis, Brazil, 2018–2019.**
(PDF)

**S3 Table. Frequency of respiratory viruses in children with and without pneumococcal carriage, Veranópolis/RS, Brazil, between 2018 and 2019.**
(PDF)

**S4 Table. Frequency of clinical symptoms and diagnosis in medical care within 14 days according to different respiratory viruses, Veranópolis/RS, Brazil, between 2018 and 2019.**
(PDF)

**S5 Table. Detection of different respiratory viruses according to seasons, Veranópolis/RS, Brazil, between 2018 and 2019.**
(PDF)

## Acknowledgments

We extend our sincere appreciation to all the staff at the Instituto Moriguchi for their cooperation and collaboration. We also thank the Veranópolis Health Department and Hospital São Peregrino Lazziozi for their assistance in collecting information from medical records. Special thanks go to the staff at the Federal University of Health Science of Porto Alegre, particularly Cristiane Bündchen, for providing statistical assistance. We would like to thank Editage (www.editage.com) for English language editing. We are grateful to all the children who participated in the study as well as to the nurses, pediatricians, and community health workers at the Public and Private Health Systems of Veranópolis.

## Author Contributions

**Conceptualization:** Kauana Pizzutti, Juliana Comerlato, Neide Maria Bruscato, Roberta Rigo Dalla Corte, Emilio Hideyuki Moriguchi, Vlademir Vicente Cantarelli.

**Data curation:** Kauana Pizzutti, Amanda Robaina, Pedro Uriel Pedrotti Vieira, Tiago Fetzner, Gabriela Rosa da Cunha, Muriel Primon de Barros, Jaqueline Verardo, João Carlos Batista Santana, Cícero Armídio Gomes Dias.

**Formal analysis:** Kauana Pizzutti, Mariana Preussler Mott, João Carlos Batista Santana, Emilio Hideyuki Moriguchi, Cícero Armídio Gomes Dias.

**Investigation:** Kauana Pizzutti, Daniele Vargas de Oliveira, Amanda Robaina, Mariana Preussler Mott, Tiago Fetzner, Gabriela Rosa da Cunha, Muriel Primon de Barros, Neide Maria Bruscato, Cícero Armídio Gomes Dias.

**Methodology:** Kauana Pizzutti, Juliana Comerlato, Daniele Vargas de Oliveira, Amanda Robaina, Mariana Preussler Mott, Pedro Uriel Pedrotti Vieira, Tiago Fetzner, Gabriela Rosa da Cunha, Muriel Primon de Barros, Jaqueline Verardo, Neide Maria Bruscato, Vlademir Vicente Cantarelli.

**Project administration:** Juliana Comerlato, Neide Maria Bruscato, Emilio Hideyuki Moriguchi, Vlademir Vicente Cantarelli, Cícero Armídio Gomes Dias.

**Resources:** João Carlos Batista Santana, Roberta Rigo Dalla Corte, Emilio Hideyuki Moriguchi, Vlademir Vicente Cantarelli, Cícero Armídio Gomes Dias.

**Software:** Jaqueline Verardo.

**Supervision:** Juliana Comerlato, Daniele Vargas de Oliveira, Mariana Preussler Mott, Neide Maria Bruscato, João Carlos Batista Santana, Roberta Rigo Dalla Corte, Emilio Hideyuki Moriguchi, Vlademir Vicente Cantarelli, Cícero Armídio Gomes Dias.

**Validation:** Kauana Pizzutti, Juliana Comerlato, Daniele Vargas de Oliveira, Amanda Robaina, Mariana Preussler Mott, Pedro Uriel Pedrotti Vieira, Jaqueline Verardo, Neide Maria Bruscato, Vlademir Vicente Cantarelli.

**Visualization:** Kauana Pizzutti, Juliana Comerlato, Amanda Robaina, Mariana Preussler Mott, Pedro Uriel Pedrotti Vieira, Tiago Fetzner, Gabriela Rosa da Cunha, Muriel Primon de Barros, Jaqueline Verardo, Neide Maria Bruscato, João Carlos Batista Santana, Roberta Rigo Dalla Corte, Vlademir Vicente Cantarelli, Cícero Armídio Gomes Dias.

**Writing – original draft:** Kauana Pizzutti, Cícero Armídio Gomes Dias.

**Writing – review & editing:** Juliana Comerlato, João Carlos Batista Santana.

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
