## [Decision Letter · Decision Letter 0]

18 Mar 2024

PONE-D-23-41647Nasopharyngeal carriage of Streptococcus pneumoniae among Brazilian children: interplay with viral co-infectionPLOS ONE

Dear Dr. Pizzutti,

Thank you for submitting your manuscript to PLOS ONE. After careful consideration, we feel that it has merit but does not fully meet PLOS ONE’s publication criteria as it currently stands. Therefore, we invite you to submit a revised version of the manuscript that addresses the points raised during the review process.

We look forward to receiving your revised manuscript.

Kind regards,

Baochuan Lin, Ph.D.

Academic Editor

PLOS ONE

Journal Requirements:

**Additional Editor Comments:**

While your paper addresses an interesting question, the reviewers stated several concerns about the presentation as well as the readability of the manuscript and did not recommend publication in its present form. In particular, reviewers voice a number of concerns regarding the data analysis, and these comments need to be addressed carefully. Please see reviewers’ insightful comments below. Personally, at a more detailed level, I find the authors should take care on how the data was calculated, for example, the prevalence of microorganism detection should based on the final number included (n=224) for analysis not the original number (n=230).

In addition, the quality of the language needs to be improved.  Please have a fluent, preferably native, English-language speaker thoroughly copyedit your manuscript for language usage, spelling, and grammar.  

Reviewers' comments:

Reviewer's Responses to Questions

**Comments to the Author**

1. Is the manuscript technically sound, and do the data support the conclusions?

Reviewer #1: Partly

Reviewer #2: Yes

Reviewer #3: Yes

2. Has the statistical analysis been performed appropriately and rigorously? 

Reviewer #1: No

Reviewer #2: Yes

Reviewer #3: Yes

3. Have the authors made all data underlying the findings in their manuscript fully available?

Reviewer #1: Yes

Reviewer #2: No

Reviewer #3: Yes

4. Is the manuscript presented in an intelligible fashion and written in standard English?

Reviewer #1: No

Reviewer #2: No

Reviewer #3: Yes

5. Review Comments to the Author

Reviewer #1: The authors present an interesting investigation into pneumococcal carriage and detection of respiratory viruses.

Abstract:

Line 26: I suggest something like ‘prerequisite’ rather than ‘compulsory’

Lines 27-28: This sentence starts by discussing the relationship between carriage and infection, but then goes on to say that the study includes asymptomatic children – which is indicative of lack of infection. This should be rephrased for clarity

Lines 42-43: from the text in the abstract, it is not clear how this work contributes to the understanding ‘between these agents and their implications for disease’ – there is no evidence of associations with virus or co-infection and outcomes?

Methods:

Line 91: What determines a ‘satisfactory’ NP swab?

Line 111: superscript the ‘TM’ on the magmax kit (it is the trademark symbol, not an abbreviation of the kit itself).

Results:

The start of this section would benefit from a little more introduction of the enrolled study population characteristics.

Line 155: ‘females’ instead of ‘girls.

Line 188: seasonality can be tested/controlled for in a multivariate analysis – this should be checked – as should they other demographic features as well as pathogen detection. Here, you can then also look at household size, school attendance etc,

Table 1: it is currently not clear, when reporting significance, what the comparator is. A more robust statistical analysis of this data is required and would strength the data in the field on the associations between demographics and pathogen detection as well as the interplay between pneumococcus and viruses within these groups.

Discussion:

Did the study team record the density of pneumococcus growth on culture plate? This could be used to look for an association between density and virus type.

The paper would benefit from more discussion of hBOV since this is a more novel finding – the discussion on HRV could be condensed – the different studies could be summarized more succinctly.

It could be useful to report the Ct values of the virus qPCR detection – if they are low viral loads (high Ct values) this would suggest either a resolving or historic infection (which I commend the authors for recognizing in the text!) so less clinically relevant. If any are higher viral loads (lower Ct) this would suggest more active infection and would be more interesting.

Reviewer #2: Major

- The manuscript needs to be reviewed for English language.

- The data statement indicates that data are available within the manuscript but only aggregate are provided and not individual-level data.

- Study design – this is a cohort study with a cross-sectional analysis and cohort analysis. It should not be described as a cross-sectional cohort study.

- Methods – more details are needed to understand who is in the study and interpret the results:

o It looks like all children in the city were eligible – how were they recruited and were siblings allowed/enrolled. If siblings were allowed, then how was that accounted for in the analysis? (it’s not clear in Table 1 if ‘more than one child in the same household’ refers to enrolled in the study or just present)

o The children were considered ‘healthy’ and are called ‘asymptomatic’ throughout the manuscript. They were excluded if they had been diagnosed with ARI or hospitalized the week before. It does not appear that they were excluded if they had respiratory symptoms before (not medically attended) or at the visit. If they were excluded then this should be stated. If they were not excluded, then it should be reported in table 1 how many were symptomatic before or at the visit and the participants should not be referred to as ‘asymptomatic’. I would also remove them from the cohort analysis since they already have symptoms or model incidence to account for this.

o The cohort analysis relies on medical records for the outcome. The authors state that they checked 15 different health systems, which seems comprehensive but without knowing this area it is hard to understand if this represents all of the health systems that they could have checked or only a subset. If it’s only a subset, then the completeness of the outcome ascertainment is called into question and it’s not clear if the lack of association is due to misclassification.

o The cohort analysis is a univariable analysis. Are there other important factors that affect healthcare seeking (e.g., SES) that may have differed between groups that should be accounted for? As few variables are presented in Table 1, it is hard to know.

o Respiratory sample collection and testing – provide brief methods (e.g., broth enrichment) so the reader can know what was done in this study without reviewing another protocol.

o Statistical analysis – the authors state that the sample size was determined based on a desired confidence interval. However, no confidence intervals are presented. Is there another paper from this study that should be cited in the Methods where the primary outcome (prevalence of PNC) is reported?

o Discussion:

Paragraph starting on line 201: Suggest refocusing to discuss that the study is consistent about an association between PNC and viruses. There was an association with hRV but it was also the most common virus detected. Similar trends were observed for all other viruses except hBOV, but with smaller sample sizes which likely contributed to the lack of statistical significance.

Paragraph starting online 219: since this is the major finding, I suggest expanding the discussion unpack a little more what you measured for both exposure and outcome:

• You only measured a subset of viruses and did not measure some major respiratory viruses (e.g., influenza, seasonal coronaviruses)

• It’s also not clear with PCR what you measured given that children were 'asymptomatic' – did you measure past infection or did you measure recent infection and the healthcare visits in the next 14 days were for the respiratory viral infection? Was this the right window for detecting pneumococcal infection after a viral infection?

• The outcome was medically attended visits and not just respiratory symptoms, which indicates a level of severity of the infection.

Minor

- Throughout the manuscript be consistent in the number of decimals presented and how numbers are presented (period vs. comma).

- Introduction, line 75: the authors state that they considered the increase in infectious disease severity when pathogens interact. This should be rephrased to more accurately reflect what was done.

- Methods

o line 92: fix typo – ‘health’ professionals

o Line 95: remove double period

- Results

o Line 150: suggest starting out by reporting the % with respiratory viruses by PNC status and providing p-value an then going into co-occurrence.

o Line 158: ‘hRV alone + co-detection’ = ‘total’ in the table. Suggest using ‘total (alone + co-detection)’ for ease of reading text.

o Line 165: suggest starting by reporting the overall % that required medical care within 14 days.

o Line 165: the authors indicate ‘co-occurrence was the most frequent factor among the 43 children who needed medical care’ but this is not what is shown in the table and incidence of medical care in this group was among the lowest. Please re-phrase.

- Figure 1

o It looks like no children were excluded for having an ARI – is that correct? Seems odd given detection of viruses.

o Box ‘refused to collect or sign ICF’ has an asterix – what is it for?

o Fix typo in box ‘children included’

- Table1

o Header row (culture isolation …) is unnecessary and can be deleted

o Clarify that PNC and respiratory virus detection columns are ‘only’

o Footnote 2 – clarify that this is ‘at least one dose’

o Footnote 4 is not necessary

o Mean (SD) is presented for age – confirm that age was normally distributed; if not, then median (IQR) would be more appropriate.

- Table 2

o Title should be changed as it does not only include children with PNC

o Rows need to be aligned

o Footnotes 2 and 3 are not necessary

- Table 3

o Suggested title: incidence of medical appointments 14 days after specimen collection among children, by colonization and respiratory virus detection

o As you are presenting incidence (row percents), the ‘no’ column is unnecessary (may cause confusion to include as Table 2 is similarly structured and presents column percents)

- Table 4

o Delete header row (culture isolation…) - it is not necessary

o Change first section results row to ‘Any documented clinical symptoms within 14 days’

o Change second section results row to ‘Any diagnosis within 14 days’

- S2 Table 1:

o Make it clearer what the top row ‘medical records’ refers to and that you are presenting row percents. The rest of the table presents column percents.

o Suggest changing ’14 days of medical care’ to ‘medical care within 14 days’

o For the individual symptoms, the percents in the ‘total’ column don’t match what is reported in table 4.

- S2 Table 2 – expand the footnote to indicate the test that yielded the p-value and what comparison is being made.

Reviewer #3: It is a very interesting article and there are very few articles in Latin America.

Some suggestions

Line 150:

It is not clear what is meant by "The prevalence of microorganism detection found in the children was...", it is suggested to use prevalence of respiratory viruses was...

Line 152:

"As shown in Table 1, children who tested positive for viral infection, with or without coexisting pneumococcus, were younger (34 vs. 40 months; p = 0.001)."

Not displayed in table 1

The lack or shortage of studies in Latin America could be included in the discussion.

Line 240-243

The high sensitivity of PCR is not a limitation, but the difficulty of interpreting the presence of viruses is.

Line 225 and 245

There is talk of high vaccination coverage; vaccination coverage could be included in the introduction in Brazil and in the state where the study is carried out.

6. PLOS authors have the option to publish the peer review history of their article (what does this mean?). If published, this will include your full peer review and any attached files.

Reviewer #1: No

Reviewer #2: No

Reviewer #3: **Yes: **María Eugenia León

---

## [Author Response · Author response to Decision Letter 0]

21 May 2024

Additional Editor Comments:

While your paper addresses an interesting question, the reviewers stated several concerns about the presentation as well as the readability of the manuscript and did not recommend publication in its present form. In particular, reviewers voice a number of concerns regarding the data analysis, and these comments need to be addressed carefully. Please see reviewers’ insightful comments below. Personally, at a more detailed level, I find the authors should take care on how the data was calculated, for example, the prevalence of microorganism detection should based on the final number included (n=224) for analysis not the original number (n=230)

We would like to thank the editors and reviewers for their dedication and thorough analysis of our manuscript. We were able to implement most of the suggestions, and in the new version the number of samples had to be changed due to a technical sample loss (from 230 to 229). As for the editor's request, we can't implement it because the number of 224 is the result of the analysis of medical records (cohort analysis) in which the sample loss was 5 children, which doesn't change the value of the prevalence of the microorganisms in the cross-sectional study of the 229 children. Information included in the Flowchart on line 157

In addition, the quality of the language needs to be improved. Please have a fluent, preferably native, English-language speaker thoroughly copyedit your manuscript for language usage, spelling, and grammar. 

The English has been revised as requested. In addition to other requests from the reviewers, lines and some references have been changed.

Reviewer #1: The authors present an interesting investigation into pneumococcal carriage and detection of respiratory viruses.

We would like to thank the reviewer for his dedication and thorough analysis of our manuscript. We were able to improve it a lot with your support

Abstract:

Line 26: I suggest something like ‘prerequisite’ rather than ‘compulsory’

It was modified as suggested. 

Lines 27-28: This sentence starts by discussing the relationship between carriage and infection, but then goes on to say that the study includes asymptomatic children – which is indicative of lack of infection. This should be rephrased for clarity

The sentence was rephrased as suggested, line 29: "in healthy children from the general population in southern Brazil"

Lines 42-43: from the text in the abstract, it is not clear how this work contributes to the understanding ‘between these agents and their implications for disease’ – there is no evidence of associations with virus or co-infection and outcomes?

The text was reviewed and modified. Line 46: "potentially pathogenic agents."

Methods:

Line 91: What determines a ‘satisfactory’ NP swab?

This sentence was rephrased, the collection guideline was included. Line 90: "Participants were recruited as volunteers through schools, telephone calls, and radio announcements in the city. All legal guardians provided written informed consent signed at the moment of specimen collection using the informed consent form (ICF). If siblings were included, the legal guardian provided consent for both children. All participants were considered healthy enough to attend school and maintain a normal routine at the time of collection. Individuals were deemed ineligible for enrollment if they had been diagnosed with acute respiratory disease or had been hospitalized in the week preceding specimen collection. Health professionals trained for the study conducted structured interviews with legal guardians to collect data on demographic characteristics, pneumococcal vaccination history, and the presence of symptoms such as cough, coryza/congestion, or sore throat in the child. Children did not undergo medical examinations."

Line 111: superscript the ‘TM’ on the magmax kit (it is the trademark symbol, not an abbreviation of the kit itself).

The alteration was done. line 121.

Results:

The start of this section would benefit from a little more introduction of the enrolled study population characteristics.

The text was reviewed and modified. Line 144: "In 2017, Veranópolis City had an estimated 781 children aged 18–59 months."

Line 155: ‘females’ instead of ‘girls.

It was modified as suggested. line 147.

Line 188: seasonality can be tested/controlled for in a multivariate analysis – this should be checked – as should they other demographic features as well as pathogen detection. Here, you can then also look at household size, school attendance etc,

The suggestion was accepted and Table 1 included all the data from the questionnaire, including seasonality. The multivariate data is in the narrative text and table 2 of the supplementary material. line 167: "Additionally, a multinomial logistic regression analysis of certain demographic data and the presence of only pneumococcal carriage, respiratory virus, and co-occurrence was conducted (S2 Table)."

Table 1: it is currently not clear, when reporting significance, what the comparator is. A more robust statistical analysis of this data is required and would strength the data in the field on the associations between demographics and pathogen detection as well as the interplay between pneumococcus and viruses within these groups.

In the multivariate (new S2 Table 1) each outcome in relation to "None", male sex in relation to female, absence of symptoms in relation to presence and each season in relation to spring.

In table 1 there is no comparator: it is the association of the variable with the microorganism variable. we have included the youngest age in the footnote of table 1 

It was modified as suggested. "⁴ Children who tested positive for virus infection and co-occurrence were younger than those that were pneumococcal carriers only"

Discussion:

Did the study team record the density of pneumococcus growth on culture plate? This could be used to look for an association between density and virus type.

Unfortunately this has not been reported as it was grown in broth and then seeded in culture 

The paper would benefit from more discussion of hBOV since this is a more novel finding – the discussion on HRV could be condensed – the different studies could be summarized more succinctly.

We understand but we chose to keep a longer discussion with hRV because now we have brought new results and there has been an association with the symptom of sore throat (line 210). A new text on bocavirus has been reformulated to emphasize the "novel finding" (line 247)

Line 210: "Children who were "too sick" were excluded from this study. The recruitment process allowed for the inclusion of children with mild respiratory symptoms, as reported by their guardians. Consequently, 108 (47.0%) and 79 (34.5%) of the children exhibited coryza/congestion and cough, respectively. Despite the frequency of these findings in our study, they were not associated with the detection of the studied viruses. In the literature, these symptoms are commonly associated with respiratory viruses. Notably, lower levels of Ct are often associated with greater severity of respiratory symptoms and mortality [40]. However, in our study, high Ct levels were observed, aligning with the clinical condition of the study group (considered healthy) and the lack of association with symptoms. Also, in five of the children, guardians reported sore throat, and in four of them, hRV was detected. Statistically significant associations were observed between any level of hRV Ct values (p<0.05) and sore throat, rather than only low hRV Ct values. Some studies have demonstrated the association of hRV with sore throat [41,42]. For instance, in a study conducted with individuals experiencing sore throat, hRV was the most commonly detected pathogen [43]. Moreover, an experimental study involving intranasal inoculation of hRV in humans observed scratchy/sore throat symptoms appearing 10 to 12 hours after induced inoculation [44]. It's important to note that as these observations about symptoms were not derived from a medical examination, some subjectivity is involved. In our study, INF and PIV were not found. The prevalence of these viruses is typically low (<3%) among asymptomatic children [15,16], as they are known to interact with hosts, producing symptoms [45,46]."

line 247: "Similarly, a Japanese study observed a positive correlation between the co-detection of pneumococcus and hBOV, where 85.7% (12/14) of children tested positive for both pathogens in otitis media fluid (MEF). However, in the nasopharynx, this correlation was not observed, as the proportion of pneumococcus was similar between groups of individuals positive and negative for hBOV. Notably, pneumococcus was the only pathogen present in all but one of the co-detections in hBOV-positive cases, whereas in the nasopharynx, it was mixed with other pathogenic bacteria [53]. This suggests that these pathogens migrate to their target sites, with the nasopharynx not being the site where the etiologic agent is identified It is essential to highlight that in this site (the nasopharynx), children without serious disease may act as antagonists, potentially suggesting protection of pneumococcus in the absence of hBOV. However, when present together, this migration may be favored. One common weakness of these studies is the small sample size, which can hinder drawing definitive conclusions about the relationship between hBOV and pneumococcus. Nevertheless, these studies collectively underscore the relevance of pneumococcus and respiratory virus co-infection in disease pathogenesis. Our study's findings, such as the absence of INF and PIV in the sampled population, likely reflect the predominance of these viruses in children with more severe symptoms. INF, in particular, is well-known for its relationship with co-infection and increased risk of subsequent severe pneumococcal disease. Further studies are needed to elucidate the mechanisms underlying the interaction between hBOV and pneumococcus."

It could be useful to report the Ct values of the virus qPCR detection – if they are low viral loads (high Ct values) this would suggest either a resolving or historic infection (which I commend the authors for recognizing in the text!) so less clinically relevant. If any are higher viral loads (lower Ct) this would suggest more active infection and would be more interesting.

It was modified as suggested. A new table 2 has been included with virus ct analysis. Line 177: 

Table 2. Frequency of clinical symptoms in children with detection of respiratory viruses by real-time PCR using the cut-off levels Cycle threshold (CT), Ct- value of < 30 (only high levels) and < 35 (any level), Veranópolis/RS, Brazil, between 2018 and 2019.

Reviewer #2: Major

We would like to thank the reviewer for his dedication and thorough analysis of our manuscript. We were able to improve it a lot with your support

- The manuscript needs to be reviewed for English language.

We carry out a new english revision

- The data statement indicates that data are available within the manuscript but only aggregate are provided and not individual-level data.

We have added an excel file to include all the results obtained for each individual child, new Supplementary Information 3 

- Study design – this is a cohort study with a cross-sectional analysis and cohort analysis. It should not be described as a cross-sectional cohort study.

It was modified as suggested. line 86:"We conducted a cross-sectional cohort study to investigate pneumococcal carriage and respiratory viruses in the nasopharynx of children aged 18–59 months."

- Methods – more details are needed to understand who is in the study and interpret the results:

o It looks like all children in the city were eligible – how were they recruited and were siblings allowed/enrolled. If siblings were allowed, then how was that accounted for in the analysis? 

The recruitment was by convenience sampling. line 90: "Participants were recruited as volunteers through schools, telephone calls, and radio announcements in the city." 

Siblings were included. line 92: "if siblings were included, the legal guardian provided consent for both children."

(it’s not clear in Table 1 if ‘more than one child in the same household’ refers to enrolled in the study or just present)

we don't exclude children from being in the same house (see inclusion and exclusion criteria). Line: "Participants were recruited as volunteers through schools, telephone calls, and radio announcements in the city. All legal guardians provided written informed consent signed at the moment of specimen collection using the informed consent form (ICF). If siblings were included, the legal guardian provided consent for both children."

o The children were considered ‘healthy’ and are called ‘asymptomatic’ throughout the manuscript. They were excluded if they had been diagnosed with ARI or hospitalized the week before. It does not appear that they were excluded if they had respiratory symptoms before (not medically attended) or at the visit. If they were excluded then this should be stated. If they were not excluded, then it should be reported in table 1 how many were symptomatic before or at the visit and the participants should not be referred to as ‘asymptomatic’.

 Was accepted and changed from "asymptomatic" to "healthy". As described in line 93: "All participants were considered healthy enough to attend school and maintain a normal routine at the time of collection. Individuals were deemed ineligible for enrollment if they had been diagnosed with acute respiratory disease or had been hospitalized in the week preceding specimen collection. Health professionals trained for the study conducted structured interviews with legal guardians to collect data on demographic characteristics, pneumococcal vaccination history, and the presence of symptoms such as cough, coryza/congestion, or sore throat in the child. Children did not undergo medical examinations."

I would also remove them from the cohort analysis since they already have symptoms or model incidence to account for this.

the 14-day analysis was to analyze whether seeking care was an aggravation of the symptoms or a possible diagnosis (of interest to the study)

o The cohort analysis relies on medical records for the outcome. The authors state that they checked 15 different health systems, which seems comprehensive but without knowing this area it is hard to understand if this represents all of the health systems that they could have checked or only a subset. If it’s only a subset, then the completeness of the outcome ascertainment is called into question and it’s not clear if the lack of association is due to misclassification.

The text was reviewed and modified. Line: 103: "covering the entire city"

o The cohort analysis is a univariable analysis. Are there other important factors that affect healthcare seeking (e.g., SES) that may have differed between groups that should be accounted for? As few variables are presented in Table 1, it is hard to know.

It was modified as suggested. A more complete table 1 (line 166) with all the data from the questionnaire and we carried out multivariate analysis (included a new table: S2 Table 1 in Supplementary Information) only for the statistically significant data from the univariate analysis.

Table 1. Demographic characteristics of the population (n = 229) according to identified microorganisms, Veranópolis/RS, Brazil, 2018–2019 

o Respiratory sample collection and testing – provide brief methods (e.g., broth enrichment) so the reader can know what was done in this study without reviewing another protocol.

The text was reviewed and modified. Line 118: " involving broth enrichment followed by growth on blood agar base."

o Statistical analysis – the authors state that the sample size was determined based on a desired confidence interval. However, no confidence intervals are presented. Is there another paper from this study that should be cited in the Methods where the primary outcome (prevalence of PNC) is reported? 

In the static item there was already the reference value line: 132 "The sample size was calculated considering a PNC prevalence of 62.3% [36]" and your s

---

## [Decision Letter · Decision Letter 1]

11 Jun 2024

PONE-D-23-41647R1Nasopharyngeal carriage of Streptococcus pneumoniae among Brazilian children: interplay with viral co-infectionPLOS ONE

Dear Dr. Pizzutti,

Thank you for submitting your manuscript to PLOS ONE. After careful consideration, we feel that it has merit but does not fully meet PLOS ONE’s publication criteria as it currently stands. Therefore, we invite you to submit a revised version of the manuscript that addresses the points raised during the review process.

Referees still have concerns with the revised manuscript. One referee does not think the explanation and revision improved the quality of the paper to the publishable standard of the journal, and the other referee also has several issues with the current revision. Please see reviewers' insightful comments below.

We look forward to receiving your revised manuscript.

Kind regards,

Baochuan Lin, Ph.D.

Academic Editor

PLOS ONE

Reviewers' comments:

Reviewer's Responses to Questions

**Comments to the Author**

1. If the authors have adequately addressed your comments raised in a previous round of review and you feel that this manuscript is now acceptable for publication, you may indicate that here to bypass the “Comments to the Author” section, enter your conflict of interest statement in the “Confidential to Editor” section, and submit your "Accept" recommendation.

Reviewer #1: (No Response)

Reviewer #2: (No Response)

Reviewer #3: All comments have been addressed

2. Is the manuscript technically sound, and do the data support the conclusions?

Reviewer #1: Partly

Reviewer #2: Yes

Reviewer #3: Yes

3. Has the statistical analysis been performed appropriately and rigorously? 

Reviewer #1: No

Reviewer #2: Yes

Reviewer #3: Yes

4. Have the authors made all data underlying the findings in their manuscript fully available?

Reviewer #1: Yes

Reviewer #2: Yes

Reviewer #3: Yes

5. Is the manuscript presented in an intelligible fashion and written in standard English?

Reviewer #1: No

Reviewer #2: Yes

Reviewer #3: Yes

6. Review Comments to the Author

Reviewer #1: Abstract, Results, lines 39-41: please consider rephrasing this as it isn’t quite clear what differences were tested for.

Abstract line 46: it is still not clear to me how these findings enhance our understanding of the relationship between these potentially pathogenic agents. There have been many other reports on the association between pneumococcus and respiratory pathogens so the authors should add additional detail to highlight was is novel about this.

Introduction – PNC and IPD are introduced twice. Please review the text – the introduction should also be reviewed to improve the flow of the messaging.

Methods/Results – were siblings controlled for in the analyses?

Results – please include a discussion of the demographic details of the 229 children enrolled that is presented in the table – at least highlight if there are any major differences between demographic groups. Please reference to the table in this section too.

Results/Conclusion – with such a high frequency of samples generating high Ct values for virus (30-35), this is most likely historic infection as the authors recognize and may not represent active infection – it is perhaps evidence of prior infection which could be skewing associations or why there isn’t higher density of pneumococcus at those sampling moments. Did you look into any association between virus and pneumococcus Ct value?

Reviewer #2: In general, the authors were very responsive to the comments and the manuscript reads much better. However, there are still a few issues to address with some of the new analyses.

Major comments:

Abstract – line 31 (of tracked version): the design should NOT be listed as a cross-sectional cohort study. This was a cohort study an cross-sectional analysis performed at baseline and a longitudinal analysis conducted with the FU data.

Methods – line 85: same comment – change to cohort study.

Methods – line 91: the authors now indicate that siblings were included but not what proportion of participants involved sibling pairs and therefore how many households are represented. If a large number of siblings were in fact enrolled, then there is clustering in the study population and this should be accounted for in the prevalence estimates and analyses.

Methods – lines 133-135: The multinomial regression needs to better described – what were the categories, what was the reference group, how were covariates selected for inclusion, etc.

Methods – line 136: the authors indicate that they used Bonferroni tests to adjust for multiple comparison but then indicate that a p<0.05 was considered statistically significant. What, if any, multiple comparisons were adjusted for and what Bonferroni-corrected p-value was considered statistically significant?

Results – line 157: the authors indicated in their response that they provided a confidence interval for the prevalence but it does not appear to be included.

Discussion – lines 249-259: the new paragraph about hBOV is somewhat confusing:

- Line 249 – does the presence of SPN as the only pathogen present refer to findings in MEF?

- Line 251 – add period after ‘identified’

- Line 251 – the sentence reads as if the children are acting as antagonists – please revise

- Line 252 – the sentence suggests protection of pneumococcus in the absence of hBOV. Don't your results suggest the opposite, that there is protection from SPN when hBOV is present? but only when it is present alone?

- Line 253 – what migration is referred to here? Migration to MEF?

- Line 259 – provide references for the well-known relationship with co-infection and increased risk of severe SPN disease

Table 1: the authors have a category called ‘self-report ethnicity’. I believe this should be ‘self-reported race’ instead. They also have an option for ‘Yellow/Indian’, which is not a term generally used – consider ‘Indigenous’ instead.

Table 2: For the p-value, it is unclear what the comparison is. Is it <30/35 vs. >30/35? if so, then the % with each symptom in the reference group needs to be shown in the table in order to interpret the results. For example, sore throat is associated with RNV<35 with 5.3% having sore throat, but is that a higher or lower % than observed in those with RNV>35?

Minor comments:

Abstract – line 37 & Results – line 157: change to 49.3% (one decimal place).

Methods – line 131-132: change ‘qualitative’ to ‘categorical’ and ‘quantitative’ to ‘continuous’

Results – line 157: add ‘of’ and ‘was’ (the prevalence OF pneumococcal carriage WAS…)

Results – line 159: italicize S. pneumoniae

Discussion – line 210: 47.2% in the table

Discussion – line 212: change ‘findings’ to ‘symptoms’

Discussion – line 224: INF and PIV are introduced as new abbreviations – spell out first

Discussion – line 268: suggest tempering language and using ‘may have prevented development of diseases’

Table 4 – suggest combining with Table 5

Table 5 – it is interesting that there isn't complete overlap in the “any documented clinical symptoms’ category and having a medical appt in Table 4. What accounts for the discrepancy?

S2 Table 1 – the current column (microorganisms) reporting n (%) for each covariate is not interpretable without the corresponding column for ‘no pathogen detected’.

S2 Table 2:

- Be consistent in the number of decimals presented in this table

- Remove the % in the top row labeled ‘medical records’ – these are row percents while the of the table presents column percents and it is confusing.

- There is a footnote outlining tests conducted and ‘* p<0.05 significant’, but it’s not clear if any comparisons are being made in this table and nothing is significant. If there are comparisons being made, please outline what they are and what the reference group is.

S2 Table 3: While the authors have indicated the statistical test being conducted, it’s still not clear what comparison is being made. For example, there is an asterix on RSV for Fall – is this comparing the distribution of seasons between those that are RSV+ and RSV-? If so, this is only interpretable if we have the distribution for the RSV- group.

Reviewer #3: (No Response)

7. PLOS authors have the option to publish the peer review history of their article (what does this mean?). If published, this will include your full peer review and any attached files.

Reviewer #1: No

Reviewer #2: No

Reviewer #3: No

---

## [Author Response · Author response to Decision Letter 1]

26 Jul 2024

We appreciate all the suggestions of reviewer 1 and 2 that, we are convinced of this, will improve our article. We examined each of the suggestions in detail and did our best to achieve a satisfactory answer. Once again, we remain at your disposal for any other suggestions.

---

## [Decision Letter · Decision Letter 2]

28 Aug 2024

PONE-D-23-41647R2Nasopharyngeal carriage of Streptococcus pneumoniae among Brazilian children: interplay with viral co-infectionPLOS ONE

Dear Dr. Pizzutti,

Thank you for submitting your manuscript to PLOS ONE. After careful consideration, we feel that it has merit but does not fully meet PLOS ONE’s publication criteria as it currently stands. Therefore, we invite you to submit a revised version of the manuscript that addresses the points raised during the review process.

Both reviewers agreed that the revised manuscript showed significant improvement, however, there are still concerns raised by the reviewers and these issues need to be addressed carefully.  Personally, I also found several issues that need to be addressed (please see specific comments below).  In addition, the quality of the language still needs to be improved. There are quite a few awkward and confusing sentences, typos throughout the manuscript.  We suggest you thoroughly copyedit your manuscript for language usage, spelling, and grammar. If you do not know anyone who can help you do this, you may wish to consider employing a professional scientific editing service.

Specific comments:

Line 39: Change hBOV to human bocavirus (hBOV)

2.     Line 126:  Change “described” to “, as described”

3.     Line 141 – 144: This is an awkward run on sentence which need rephrasing for clarity.

4.     Line 151:  If 6 were excluded, this should be 223. In figure 1, 5 were excluded, which would be 224. Please make sure that the numbers are consistent throughout the manuscript.

5.     Line 160 – 161: Change “… for and respiratory viruses 49.3% (113/229, CI95%: 42.9% - 55.8%).” To “…and 49.3% (113/229, CI95%: 42.9% - 55.8%) for respiratory viruses.”

6.     Line 163:  Delete “for both S. pneumoniae and any virus”

7.     Table 1:  Please include Female number for ease of reading.

8.     Line 172:  Change “associated with co-occurrence” to “associated with higher rate of co-occurrence”

9.     Line 181 – 183:  This is an awkward sentence, please rephrase for clarity.

10.  Table 3:  Do the authors of the information of co-detection of viruses? It maybe helpful to have more details instead of just total number, for example hRV + ADV or hRV + RSV?

11.  Line 204: Change “demonstrate” to “revealed”

12.  Line 206 – 207:  This is an awkward sentence, not sure what the authors wish to convey.

13.  Line 220:  Not sure the significance and/or relevance of this sentence. Please clarify.

14.  Line 256: This sentence missed a period at the end.

15.  Line 256 – 257: This sentence needs rephrasing for clarity.

We look forward to receiving your revised manuscript.

Kind regards,

Baochuan Lin, Ph.D.

Academic Editor

PLOS ONE

Journal Requirements:

Reviewers' comments:

Reviewer's Responses to Questions

**Comments to the Author**

1. If the authors have adequately addressed your comments raised in a previous round of review and you feel that this manuscript is now acceptable for publication, you may indicate that here to bypass the “Comments to the Author” section, enter your conflict of interest statement in the “Confidential to Editor” section, and submit your "Accept" recommendation.

Reviewer #1: (No Response)

Reviewer #2: (No Response)

2. Is the manuscript technically sound, and do the data support the conclusions?

Reviewer #1: Yes

Reviewer #2: Yes

3. Has the statistical analysis been performed appropriately and rigorously? 

Reviewer #1: I Don't Know

Reviewer #2: Yes

4. Have the authors made all data underlying the findings in their manuscript fully available?

Reviewer #1: Yes

Reviewer #2: Yes

5. Is the manuscript presented in an intelligible fashion and written in standard English?

Reviewer #1: Yes

Reviewer #2: Yes

6. Review Comments to the Author

Reviewer #1: I commend the authors for the work they have put in to further update their manuscript. I only have a couple more comments:

I believe line 38 where it is talking about respiratory viruses in carriers is also missing a ‘versus non-carriers’. The ‘versus non-carriers’ could move from line 39 to 38 to introduce this comparison and carry that through the subsequent comparisons.

It is still not entirely clear how the results enhance our understanding (line 47). That sentence could since the authors have added a new sentence about the study after that.

Line 165 – it is not clear in the text to what the p-value represents.

Reviewer #2: The authors have adequately addressed most comments. There are two outstanding comments:

Table 2: while the authors have clarified that they performed the analysis using two different cutoffs, the table only presents one of the proportions from the comparison being made. Presumably, the comparison being made and referenced by the p-value is the % symptomatic with virus <30 (or <35) compared to the % symptomatic with virus >=30 (or >=35). Given that only RNV is significant at ct<35, I suggest presenting the comparison of <35 vs. >=35 in Table 2 and creating a supplemental table to show the results for <30 vs. >=30 and just refer to the results being consistent at this lower ct cutoff in the text. Also, combine the first two footnotes (e.g., * p<0.05 significant by chi-square or Fisher’s exact test) – the same should be done for Table 1 as well.

Methods - line 138-141: Can the authors clarify if the age comparison using the Kruskal-Wallis was followed by post-hoc Dunn’s test with Bonferroni adjustment. The sentences should be clarified to indicate what was done (e.g., The association between pneumococcal carriage and age was conducted using a Kruskal-Wallis test and followed with post-hoc Dunn’s pairwise tests and Bonferroni correction. The association of pneumococcal carriage and demographic data other than age, symptoms,…).

Additional minor comments:

Abstract – line 31 (tracked version): modify sentence to indicate 229 nasopharyngeal samples collected from children aged 18-59 months at baseline were included.

Methods – line 88-101: modify sentence in line 88 to delete ‘and cross-sectional analysis’. Modify line 98 to indicate ‘at enrollment, health professionals trained for the study collected a nasopharyngeal specimen and conducted structure…..’.

Methods – line 127-129: move new text to Statistical Analysis section and describe this analysis (chi-square or Fisher’s exact test).

Methods – line 141-144: revise the sentence regarding multinomial regression for English (e.g., The multinomial logistic regression included four outcome categories including ‘agent identified’ (reference group), ‘pneumococcal carriage only’, ‘ respiratory virus detection only’, and ‘co-occurrence’. Exposure variables included age in months, male sex, does not sleep alone, season, and any illness symptoms at enrollment.)

Methods – line 135: add abbreviation for confidence level (CI) as it is used in the results.

Results – line 135: clarify that it is ‘PNC prevalence at enrollment’

Results – line 164: fix grammar leading to prevalence for respiratory viruses

Results – line 165: a p-value is provided for the % with respiratory viruses among those with PNC but it is not interpretable unless the % among those without PNC is provided – either remove or provide the % (e.g., p=0.033 compared to X% among those without PNC).

Results – line 166: italicize S. pneumoniae

Discussion – line 261 – put back in the period after ‘hBOV alone’.

References - italicize S. pneumoniae throughout

7. PLOS authors have the option to publish the peer review history of their article (what does this mean?). If published, this will include your full peer review and any attached files.

Reviewer #1: No

Reviewer #2: No

---

## [Author Response · Author response to Decision Letter 2]

14 Oct 2024

The authors of this manuscript would like to thank the editors and reviewers of the journal Plos One for the excellent progress. We believe that we have made a remarkable improvement. The indication of Editage was also very important in this progress towards publication. Thanks to everyone involved. Here are the changes to the manuscript detailed in this document:

Specific comments:

Line 39: Change hBOV to human bocavirus (hBOV)

It was modified as suggested. 

2. Line 126: Change “described” to “, as described”

It was modified as suggested. 

3. Line 141 – 144: This is an awkward run on sentence which need rephrasing for clarity.

It was modified as suggested.line 129: "The association between pneumococcal carriage and/or respiratory virus detection and age was assessed using the Kruskal–Wallis test followed by post-hoc Dunn’s pairwise tests and Bonferroni correction."

4. Line 151: If 6 were excluded, this should be 223. In figure 1, 5 were excluded, which would be 224. Please make sure that the numbers are consistent throughout the manuscript.

we were wrong to write 6 because there were 5 children excluded from the medical records

5. Line 160 – 161: Change “… for and respiratory viruses 49.3% (113/229, CI95%: 42.9% - 55.8%).” To “…and 49.3% (113/229, CI95%: 42.9% - 55.8%) for respiratory viruses.”

It was modified as suggested.

6. Line 163: Delete “for both S. pneumoniae and any virus”

It was modified as suggested.

7. Table 1: Please include Female number for ease of reading.

It was modified as suggested: "Table 1" 

Female Sex

114 (49.8%)

36 (54.5%)

16 (48.5%)

 29 (36.2%)

33 (66%)

-

8. Line 172: Change “associated with co-occurrence” to “associated with higher rate of co-occurrence”

It was modified as suggested.

9. Line 181 – 183: This is an awkward sentence, please rephrase for clarity.

It was modified as suggested. line 173: "Various viruses were detected in children with pneumococcal infections. A relationship was observed between total hRV (alone + co-detection) 39% (57/146, p = 0.011) and the detection of any respiratory virus 54.8% (80/146, p = 0.040) with pneumococcal carriage (Table 3). Additionally, the presence of hBOV alone was associated with the absence of pneumococcal carriage (10.8%; p = 0.016). All samples tested negative for INF A and B, as well as for hPIV 1, 2, and 3."

10. Table 3: Do the authors of the information of co-detection of viruses? It maybe helpful to have more details instead of just total number, for example hRV + ADV or hRV + RSV?

We agree with the request and have included a description of the “total” (viral co-occurrence), but the table was too long. That's why we've built an additional Table S2 in Supplementary Information 2:

Table S2. Frequency of respiratory viruses in children with and without pneumococcal carriage, Veranópolis/RS, Brazil, between 2018 and 2019.

Respiratory virus detection

Pneumococcal carriage

p-value

Yes (n=146)

No (n=83)

hRV

Alone (n = 51)

38 (26%)

13 (15.7%)

0.100

Total¹ (n = 75)

57 (39%)

18 (21.7%)

0.011

Virus Co-occurrence ( n = 24) 

hRV + BOV (n = 1)

hRV + ADV (n = 4)

hRV + ADV + BOV (n = 2)

hRV + ADV + MPV (n = 2)

hRV + ADV + RSV (n = 1)

hRV + MPV (n = 5)

hRV + MPV + BOV (n = 4)

hRV + RSV (n = 2)

hRV + RSV + MPV (n = 1)

hRV + RSV + BOV (n = 1)

hRV + ADV + MPV + RSV (n = 1)

ADV

Alone (n = 9)

6 (4.1%)

3 (3.6%)

1.000

Total¹ (n = 23)

18 (12.3%)

5 (6%)

0.195

Virus Co-occurrence ( n = 14) 

hRV + ADV (n = 4)

hRV + ADV + BOV (n = 2)

hRV + ADV + MPV (n = 2)

hRV + ADV + RSV (n = 1)

hRV + ADV + MPV + RSV (n = 1)

ADV + BOV (n = 2)

ADV + BOV + RSV (n = 1)

ADV + BOV + RSV + MPV (n = 1)

hBOV

Alone (n = 13)

4 (2.7%)

9 (10.8%)

0.016 

Total¹ (n = 27)

16 (11%)

11 (13.3%)

0.761

Virus Co-occurrence ( n = 14) 

hRV + BOV (n = 1)

hRV + ADV + BOV (n = 2)

hRV + MPV + BOV (n = 4)

hRV + RSV + BOV (n = 1)

ADV + BOV (n = 2)

ADV + BOV + RSV (n = 1)

ADV + BOV + RSV + MPV (n = 1)

BOV + MPV (n = 1)

BOV + RSV (n = 1)

RSV

Alone (n = 8)

7 (4.8%)

1(1.2%)

0.264

Total¹ (n = 17)

14 (9.6%)

3 (3.6%)

0.163

Virus Co-occurrence ( n = 9)

hRV + RSV (n = 2)

hRV + RSV + MPV (n = 1)

hRV + RSV + BOV (n = 1)

hRV + ADV + RSV (n = 1)

hRV + ADV + MPV + RSV (n = 1)

ADV + BOV + RSV (n = 1)

ADV + BOV + RSV + MPV (n = 1) 

BOV + RSV (n = 1)

MPV

Alone (n = 2)

1 (0.7%)

1 (1.2%)

1.000

Total¹ (n = 17)

Virus Co-occurrence ( n = 15)

13 (8.9%)

4 (4.8%)

0.384

hRV + ADV + MPV (n = 2)

hRV + MPV (n = 5)

hRV + MPV + BOV (n = 4)

hRV + RSV + MPV (n = 1)

hRV + ADV + MPV + RSV (n = 1)

ADV + BOV + RSV + MPV (n = 1)

BOV + MPV (n = 1)

Any respiratory virus (n=113)

80 (54.8%)

33 (39.7%)

0.040

The chi-square or Fisher’s exact test

p < 0.05 significant

¹ Total = alone + co-detection with another respiratory virus

11. Line 204: Change “demonstrate” to “revealed”

It was modified as suggested.

12. Line 206 – 207: This is an awkward sentence, not sure what the authors wish to convey.

It was modified as suggested. line 184: "However, no statistically significant correlation was found between pneumococcal carriage, respiratory virus detection, or co-occurrence in the medical appointments of the participants 14 days after specimen collection (Table 4)"

13. Line 220: Not sure the significance and/or relevance of this sentence. Please clarify.

It was modified as suggested. line 168: "The only symptom associated with the occurrence of any virus was sore throat with any hRV Ct value (p < 0.05)."

14. Line 256: This sentence missed a period at the end.

It was modified as suggested.

15. Line 256 – 257: This sentence needs rephrasing for clarity.

It was modified as suggested. line 246: "However, interpretation of the results of both studies was limited by the small number of patients included."

Reviewer #1: I commend the authors for the work they have put in to further update their manuscript. I only have a couple more comments:

I believe line 38 where it is talking about respiratory viruses in carriers is also missing a ‘versus non-carriers’. The ‘versus non-carriers’ could move from line 39 to 38 to introduce this comparison and carry that through the subsequent comparisons.

It was modified as suggested.

It is still not entirely clear how the results enhance our understanding (line 47). That sentence could since the authors have added a new sentence about the study after that.

We thank the reviewer's comment and deleted the sentence: "These findings enhance our understanding of the relationship between these potentially pathogenic agents."

Line 165 – it is not clear in the text to what the p-value represents. 

It was modified as suggested. line 154: "As shown in Table 1, children who tested positive for viral infection, with or without the co-occurrence of pneumococcus, were younger than pneumococcal carriers (31 and 33 months vs. 42 months; p < 0.001)."

Reviewer #2: The authors have adequately addressed most comments. There are two outstanding comments:

Table 2: while the authors have clarified that they performed the analysis using two different cutoffs, the table only presents one of the proportions from the comparison being made. Presumably, the comparison being made and referenced by the p-value is the % symptomatic with virus <30 (or <35) compared to the % symptomatic with virus >=30 (or >=35). Given that only RNV is significant at ct<35, I suggest presenting the comparison of <35 vs. >=35 in Table 2 and creating a supplemental table to show the results for <30 vs. >=30 and just refer to the results being consistent at this lower ct cutoff in the text. 

Following the suggestions of the reviewers, we explored the cutoff values and the associations with symptoms. We can present only results of the Ct 35, as suggested (this was the original version of the table) and create a new supplementary table. However, we consider that Table 2, with different cutoffs, is necessary at this point.

Also, combine the first two footnotes (e.g., * p<0.05 significant by chi-square or Fisher’s exact test) – the same should be done for Table 1 as well. It was modified as suggested. 

Methods - line 138-141: Can the authors clarify if the age comparison using the Kruskal-Wallis was followed by post-hoc Dunn’s test with Bonferroni adjustment. The sentences should be clarified to indicate what was done (e.g., The association between pneumococcal carriage and age was conducted using a Kruskal-Wallis test and followed with post-hoc Dunn’s pairwise tests and Bonferroni correction. The association of pneumococcal carriage and demographic data other than age, symptoms,…).

It was modified as suggested. line:129 "The association between pneumococcal carriage and/or respiratory virus detection and age was assessed using the Kruskal–Wallis test followed by post-hoc Dunn’s pairwise tests and Bonferroni correction."

Additional minor comments:

Abstract – line 31 (tracked version): modify sentence to indicate 229 nasopharyngeal samples collected from children aged 18-59 months at baseline were included.

It was modified as suggested

Methods – line 88-101: modify sentence in line 88 to delete ‘and cross-sectional analysis’. 

It was modified as suggested

Modify line 98 to indicate ‘at enrollment, health professionals trained for the study collected a nasopharyngeal specimen and conducted structure…..’.

It was modified as suggested

Methods – line 127-129: move new text to Statistical Analysis section and describe this analysis (chi-square or Fisher’s exact test).

It was modified as suggested.line 136: "A chi-square or Fisher's exact test was carried out on the symptoms at the time of collection, and each virus was investigated using two cut-off points: Ct <30 (high levels only) and <35 (any level); Ct >35 was considered negative."

Methods – line 141-144: revise the sentence regarding multinomial regression for English (e.g., The multinomial logistic regression included four outcome categories including ‘agent identified’ (reference group), ‘pneumococcal carriage only’, ‘ respiratory virus detection only’, and ‘co-occurrence’. Exposure variables included age in months, male sex, does not sleep alone, season, and any illness symptoms at enrollment.)

It was modified as suggested. line 33: "The multinomial logistic regression included four outcome categories including ‘agent identified’ (reference group), ‘pneumococcal carriage only, ‘ ‘respiratory virus detection only,’ and ‘co-occurrence.’ The exposure variables included age (months), male sex, lack of sleep, season, and any illness symptoms at enrollment. "

Methods – line 135: add abbreviation for confidence level (CI) as it is used in the results.

It was modified as suggested. 

Results – line 135: clarify that it is ‘PNC prevalence at enrollment’

It was modified as suggested: "The sample size was calculated considering the Brazilian PNC prevalence of 62.3%"

Results – line 164: fix grammar leading to prevalence for respiratory viruses

It was modified as suggested. 

Results – line 165: a p-value is provided for the % with respiratory viruses among those with PNC but it is not interpretable unless the % among those without PNC is provided – either remove or provide the % (e.g., p=0.033 compared to X% among those without PNC).

It was modified as suggested. "The prevalence of pneumococcal carriage was 63.7% (146/229, CI 95%: 57.4–69.8%) and 49.3% (113/229, CI 95%: 42.9–55.8%) for respiratory viruses. Respiratory viruses were more frequently found among pneumococcal carriers (54.4% (80/146) vs. non-carriers [39.7% (33/83); p = 0.033] ). CI95%: 46.7–62.7%)."

Results – line 166: italicize S. pneumoniae

It was modified as suggested. 

Discussion – line 261 – put back in the period after ‘hBOV alone’.

It was modified as suggested. 

References - italicize S. pneumoniae throughout

It was modified as suggested.

---

## [Decision Letter · Decision Letter 3]

5 Nov 2024

PONE-D-23-41647R3Nasopharyngeal carriage of Streptococcus pneumoniae among Brazilian children: interplay with viral co-infectionPLOS ONE

Dear Dr. Pizzutti,

Thank you for submitting your manuscript to PLOS ONE. After careful consideration, we feel that it has merit but does not fully meet PLOS ONE’s publication criteria as it currently stands. Therefore, we invite you to submit a revised version of the manuscript that addresses the points raised during the review process.

The revised manuscript showed significant improvement, however, the reviewer still have one issue that need to be addressed. In addition, on line 71, I suggest that you change  "there information..." to "the information..."

We look forward to receiving your revised manuscript.

Kind regards,

Baochuan Lin, Ph.D.

Academic Editor

PLOS ONE

Journal Requirements:

Reviewers' comments:

Reviewer's Responses to Questions

**Comments to the Author**

1. If the authors have adequately addressed your comments raised in a previous round of review and you feel that this manuscript is now acceptable for publication, you may indicate that here to bypass the “Comments to the Author” section, enter your conflict of interest statement in the “Confidential to Editor” section, and submit your "Accept" recommendation.

Reviewer #2: (No Response)

2. Is the manuscript technically sound, and do the data support the conclusions?

Reviewer #2: Yes

3. Has the statistical analysis been performed appropriately and rigorously? 

Reviewer #2: Yes

4. Have the authors made all data underlying the findings in their manuscript fully available?

Reviewer #2: Yes

5. Is the manuscript presented in an intelligible fashion and written in standard English?

Reviewer #2: Yes

6. Review Comments to the Author

Reviewer #2: I thank the authors for addressing the comments. The manuscript is greatly improved. I have just one remaining comment regarding Table 2 that should be addressed before the manuscript can be accepted. The authors present two cutoffs (which is fine) but do not present the comparison group (which I believe is >35 for both for both cutoffs). Without presenting the results for >35, it is not possible to interpret the % provided. For example, 5.3% of RNV with >35 have a sore throat, but what is that being compared to? is that higher or lower than among those with >35? A column for >35 should be added for each pathogen.

7. PLOS authors have the option to publish the peer review history of their article (what does this mean?). If published, this will include your full peer review and any attached files.

Reviewer #2: No

---

## [Author Response · Author response to Decision Letter 3]

22 Nov 2024

The authors of this manuscript would like to thank the editors and reviewers of the journal Plos One for the excellent progress. Thanks to everyone involved. Here are the changes to the manuscript detailed in this document:

6. Review Comments to the Author

Reviewer #2: I thank the authors for addressing the comments. The manuscript is greatly improved. I have just one remaining comment regarding Table 2 that should be addressed before the manuscript can be accepted. The authors present two cutoffs (which is fine) but do not present the comparison group (which I believe is >35 for both for both cutoffs). Without presenting the results for >35, it is not possible to interpret the % provided. For example, 5.3% of RNV with >35 have a sore throat, but what is that being compared to? is that higher or lower than among those with >35? A column for >35 should be added for each pathogen.

It was modified as suggested: "Table 2" 

Table 2. Frequency of clinical symptoms in children with detection of respiratory viruses by real-time PCR using the cut-off levels Cycle threshold (CT), Ct- value of < 30 (only high levels) and < 35 (any level), Veranópolis/RS, Brazil, between 2018 and 2019.

Total

RNV

ADV

BOV

RSV

MPV

Any respiratory virus

Cycle Threshold (Ct)

<30

<35

> 35

<30

<35

> 35

<30

<35

>35

<30

<35

>35

<30

<35

>35

<30

<35

>35

n, (%)

229

33 (14.4)

75 (32.8)

154 (67.2)

9 (3.9)

23 (10)

206 (90)

6 (2.6)

27 (11.8)

202 (88.2)

3 (1.3)

17 (7.4)

212 (92.6)

2 (0.9)

17 (7.4)

212 (92.6)

47 (20.5)

113 (49.3)

116 (50.7)

Cough

79 (34.5)

11 (33.3)

26 (34.7)

53 (34.4)

2 (22.2)

5 (21.7)

74 (35.9)

0 

(0)

8 (29.6)

71 (35.1)

0

 (0)

4 (23.5)

75 (35.4)

0 

(0)

5 (29.4)

74 (34.9)

13 (27.7)

38 

(33.6)

41 (35.3)

Nasal congestion/coryza

108 (47.2)

17 (51.5)

35 (46.7)

73 (47.4)

3 (33.3)

10 (43.5)

98 (47.6)

1 (16.7)

10 (37)

98 (48.5)

0 

(0)

5 (29.4)

103 (48.6)

1 (50)

7 (41.2)

101 (47.6)

21 (44.7)

48 

(42.5)

60 (51.7)

Sore throat

5 (2.2)

2 

(6.1)

4 (5.3)

1 (0.6)

0

 (0)

0 

(0)

5 (2.4)

0 

(0)

0 

(0)

5 (2.5)

0 

(0)

0 

(0)

5 (2.4)

0 

(0)

0

(0)

5 (2.4)

2 (4.3)

4 

(3.5)

1 (0.9)

p<0.05 significant by Chi-square or Fisher’s exact test

ct >35 was negative

---

## [Decision Letter · Decision Letter 4]

11 Dec 2024

Nasopharyngeal carriage of Streptococcus pneumoniae among Brazilian children: interplay with viral co-infection

PONE-D-23-41647R4

Dear Dr. Kauana Pizzutti,

We’re pleased to inform you that your manuscript has been judged scientifically suitable for publication and will be formally accepted for publication once it meets all outstanding technical requirements.

Kind regards,

Jairam Meena, Ph.D

Academic Editor

PLOS ONE

Additional Editor Comments (optional):

Reviewers' comments:

Reviewer's Responses to Questions

**Comments to the Author**

1. If the authors have adequately addressed your comments raised in a previous round of review and you feel that this manuscript is now acceptable for publication, you may indicate that here to bypass the “Comments to the Author” section, enter your conflict of interest statement in the “Confidential to Editor” section, and submit your "Accept" recommendation.

Reviewer #2: All comments have been addressed

2. Is the manuscript technically sound, and do the data support the conclusions?

Reviewer #2: Yes

3. Has the statistical analysis been performed appropriately and rigorously? 

Reviewer #2: Yes

4. Have the authors made all data underlying the findings in their manuscript fully available?

Reviewer #2: Yes

5. Is the manuscript presented in an intelligible fashion and written in standard English?

Reviewer #2: Yes

6. Review Comments to the Author

Reviewer #2: No further comments - the authors have addressed all comments and I am happy to approve for publication.

7. PLOS authors have the option to publish the peer review history of their article (what does this mean?). If published, this will include your full peer review and any attached files.

Reviewer #2: No

---

## [Editor Report · Acceptance letter]

19 Dec 2024

PONE-D-23-41647R4 

PLOS ONE

Dear Dr. Pizzutti, 

I'm pleased to inform you that your manuscript has been deemed suitable for publication in PLOS ONE. Congratulations! Your manuscript is now being handed over to our production team.

Kind regards, 

on behalf of

Dr. Jairam Meena 

Academic Editor

PLOS ONE